# Predicting the Occurrence of Advanced Schistosomiasis Based on FISHER Discriminant Analysis of Hematological Biomarkers

**DOI:** 10.3390/pathogens11091004

**Published:** 2022-09-03

**Authors:** Fei Hu, Fan Yang, Huiqun Xie, Zhulu Gao, Jing Xu, An Ning, Shuying Xie

**Affiliations:** 1Jiangxi Provincial Institute of Parasitic Diseases, Nanchang 330096, China; 2National Institute of Parasitic Diseases, Chinese Center for Disease Control and Prevention, WHO Collaborating Centre for Tropical Diseases, Chinese Center for Tropical Disease Research, Shanghai 200025, China

**Keywords:** advanced schistosomiasis, biomarkers, fisher discriminant analysis, predicting, prediction model

## Abstract

We established a model that predicts the possibility of chronic schistosomiasis (CS) patients developing into advanced schistosomiasis (AS) patients using special biomarkers that were detected in human peripheral blood. Blood biomarkers from two cohorts (132 CS cases and 139 AS cases) were examined and data were collected and analyzed by univariate and multivariate logistic regression analysis. Fisher discriminant analysis (FDA) for advanced schistosomiasis was established based on specific predictive diagnostic indicators and its accuracy was assessed using data of 109 CS. The results showed that seven indicators including HGB, MON, GLB, GGT, APTT, VIII, and Fbg match the model. The accuracy of the FDA was assessed by cross-validation, and 86.7% of the participants were correctly classified into AS and CS groups. Blood biomarker data from 109 CS patients were converted into the discriminant function to determine the possibility of occurrence of AS. The results demonstrated that the possibility of occurrence of AS and CS was 62.1% and 89.0%, respectively, and the accuracy of the established model was 81.4%. Evidence displayed that Fisher discriminant analysis is a reliable predictive model in the clinical field. It’s an important guide to effectively control the occurrence of AS and lay a solid foundation for achieving the goal of schistosomiasis elimination.

## 1. Introduction

Schistosomiasis is a zoonotic parasitic disease that is caused by the trematode flukes of *Schistosoma* spp., mainly endemic in 78 countries and territories of Asia, Africa, and South America [1], which is still one of the most serious global public health problems [2]. *Schistosomiasis japonica* is widely distributed in 12 provinces (cities and autonomous regions) in China [3], posing a serious threat to human health and socio-economic development [4,5]. Although schistosomiasis has been in a state of low endemicity in China [6], new challenges have emerged. Some patients with chronic schistosomiasis who have not received timely and thorough treatment are gradually developing into advanced schistosomiasis [7,8]. A total of 30,170 cases of advanced schistosomiasis still exist in China [9], while some of them are discovered and reported in areas where the transmission of schistosomiasis has been interrupted [10].

Clinical presentation varies in chronic schistosomiasis cases, but many cases present mild symptoms or an absence of symptoms thus leading to misdiagnosis or a lack of treatment. Advanced schistosomiasis is mainly characterized by liver and spleen lesions, such as periportal hepatic fibrosis, portal hypertension, spleen enlargement, and congestion [11]. Tissue fibrosis that is caused by egg deposition is the most serious outcome of schistosome infection. The long course of the disease, high treatment costs, and poor prognosis impose huge psychological and economic burdens on patients and their families [12,13,14]. Schistosomiasis is an immune disease, and its pathological basis is the immune response of the host to schistosomes and eggs [15]. The schistosome eggs that are produced by adult worms in the portal venous system enter the liver with the bloodstream. Due to the large diameter of the eggs, they stay in front of the hepatic sinus to block the blood vessels and release egg antigens to sensitize delayed allergic T-lymphocytes, resulting in delayed sensitization. Allergy-sensitized T-lymphocytes produce a series of lymphokines attracting inflammatory cells (including macrophages, lymphocytes, eosinophils, etc.) to gather around the eggs, causing a series of inflammation to form egg granulomas. In the egg granulomas, macrophages release pro-fibrotic cytokines to activate hepatic stellate cells (HSCs) to transform into myofibroblasts, which produce a large amount of extracellular matrix (ECM) and secrete pro-fibrotic factors [16,17,18], resulting in an imbalance between fiber generation and degradation. Without effective intervention, fibrogenesis progresses further and eventually develops into advanced schistosomiasis. Complete recovery from advanced schistosomiasis is difficult to achieve, therefore, with the increasing disease burden of advanced schistosomiasis cases in the entire schistosomiasis disease spectrum, it is necessary to provide effective preventive strategies for early detection, early diagnosis, and early treatment, which will benefit the control and elimination of schistosomiasis and decrease the burden of the schistosomiasis population. However, few studies that are related to this have been conducted so far.

Currently, the main diagnostic methods for schistosomiasis liver fibrosis are: (1) histological examination, which is a reliable method for diagnosing liver fibrosis, but has great trauma to the tissue, and the sampling site cannot reflect the overall condition of the patient’s liver; (2) imaging methods, ultrasonography is the most widely used method for diagnosing schistosomiasis liver disease or assessing curative effectiveness based on grading of liver fibrosis; (3) detecting biomarkers of serum or plasma, including extracellular matrix components, degradation products, and metabolism of enzymes and cytokines, etc., which is readily assayed and non-invasive [19,20], but its specificity needs to be assessed as liver parenchymal damage that is caused by schistosomiasis, chronic hepatitis B, and cirrhosis may lead to abnormal synthesis of coagulation factors, anticoagulant substances, and fibrinolytic substances [20]. As the liver is the main site for the synthesis of coagulation factors, anticoagulant substances (such as AT-III, PC, PS) and fibrinolytic substances (such as PLG) in the blood, liver parenchymal damage may lead to abnormal synthesis of those factors or substances [20]. In our previous study, we found that more than 50% of patients with advanced schistosomiasis had abnormal levels of D-dimer in plasma, and there were weak positive correlations between the concentration of D-dimer and the grade of hepatic fibrosis [21]. The result indicated that some special biomarkers that were detected in peripheral blood might be used for predicting the occurrence of advanced schistosomiasis.

This study was designed to analyze and compare peripheral blood-related metabolic indicators in patients with chronic schistosomiasis and advanced schistosomiasis to select specific predictive diagnostic indicators and establish a Fisher discriminant analysis for advanced *Schistosomiasis japonica*, to achieve early detection and early intervention, and effectively reduce the new cases of advanced schistosomiasis japonica.

## 2. Results

### 2.1. General Information of Patients

A total of 139 AS patients and 132 CS patients were included in this study. The ratios of males and females participating in this study were 1:0.76 and 1:1.03 for AS and CS patients, respectively (*χ*^2^ = 1.567, *p* = 0.211). The average age of male AS and CS patients was 64.1 years and 61.8 years, and females with AS and CS aged 57.8 and 59.9 years old, respectively, with no statistically significant difference that was detected between the two groups with the same gender (*t*_Male_ = 2.401, *t*_Female_ = 1.484, *p*_all_ > 0.05). The AS and CS groups were matched for gender and age (Table 1).

### 2.2. Data Preprocessing

Taking AS and CS patients as state variables, and the detection value of blood biomarkers as test variables, the ROC curve coordinates corresponding to the sensitivity and specificity were used to calculate the maximum Youden Index, which is the optimal diagnostic cut-off value of biomarkers for two groups of patients (Appendix A). As 8 of the 30 blood biomarkers including WBC, PLT, EOS, ALT, TP, ALB, AFP, and IV-C could not calculate the optimal diagnostic cut-off value, they were discarded in further analysis (Table 2).

### 2.3. Univariate Analysis

After converting the quantitative values of the selected 22 blood biomarkers into qualitative 0 and 1 based on the optimal diagnostic cutoff value, the Chi-square test was performed with six biomarkers including SJ and five hepatitis B items. The results showed that except for seven biomarkers such as SJ, RBC, HbsAg, HbsAb, HbeAg, HbeAb, and HbcAb, without statistical significance being detected, other biomarkers presented significant differences between the AS group and CS group (Table 3). However, due to the *p*-value of RBC and HbcAb < 0.2, we still included it in the multivariate analysis according to the initial statistical constraints.

### 2.4. Multivariate Logistic Regression Analysis

The 23 blood biomarkers with *p* < 0.2 in the univariate analysis were subjected to multivariate analysis, and finally, nine blood biomarkers including HGB, LYM, MON, DBiL, GLB, GGT, APTT, Fbg, and VIII were entered into the following analysis (Table 4).

### 2.5. Establishment of Classification Model

The AS group and the CS group were distinguished according to the blood biomarkers, and the FDA was constructed based on the nine biomarkers that were screened by multivariate analysis. According to the results of the statistical analysis, seven variables with statistical significance were screened out: HGB (X_1_), MON (X_2_), GLB (X_3_), GGT (X_4_), APTT (X_5_), VIII(X_6_), and Fbg (X_7_), thus the following discriminant function was obtained (Wilks’ lambda = 0.624, χ^2^ = 125.033, df = 7, *p* = 0.000):C_AS_ = 0.923X_1_ + 3.058X_2_ + 2.672X_3_ + 2.694X_4_ + 4.364X_5_ + 2.226X_6_ + 7.744X_7_ − 8.211
C_CS_ = 1.843X_1_ + 1.930X_2_ + 1.002X_3_ + 1.586X_4_ + 2.893X_5_ + 2.863X_6_ + 8.875X_7_ − 7.621

Then, the quantitative values of the selected seven blood biomarkers of patients were converted into qualitative 0 or 1 based on the optimal diagnostic cutoff value, which was substituted into the C_AS_ and C_CS_ equations, and the values of these two equations were calculated, respectively. By comparing the C_AS_ and C_CS_ values, the patients were classified according to the following principles: if C_AS_ > C_CS_, it was determined to be AS patients; otherwise they were judged as CS patients. The accuracy of the FDA was assessed by cross-validation. The results showed that 86.7% of the participants were correctly classified into AS group and CS group (Table 5).

### 2.6. Prediction Accuracy of the FDA Model

We further replaced the discriminant functions with blood biomarker data of 109 CS patients and calculated the values of C_AS_ and C_CS_, respectively. By comparing the C_AS_ and C_CS_ values, the CS patients who were not included in the statistical analysis were classified according to the set principles. A return workup was also conducted in 2020 to determine whether these patients have developed into AS cases. A total of seven CS patients were lost in this visit. The results showed that among the 29 patients who were judged to be AS cases by the discriminant function, 18 patients eventually developed into AS cases, with a coincidence rate of 62.1%. Among the 73 patients who were judged to be CS cases, eight patients eventually developed into AS cases, accounting for the number of visitors. The overall coincidence rate was 81.4% (Table 6).

## 3. Discussion

Schistosomiasis is a serious parasitic disease that is characterized by immunopathological damage. The deposition of eggs in the liver induces an immune response leading to liver inflammation and fibrosis [22,23]. If the process of liver fibrosis for patients cannot be effectively controlled, it will develop into advanced schistosomiasis, manifesting portal hypertension, splenomegaly, and complications including hypersplenism, esophageal variceal bleeding, and hepatic encephalopathy, etc. [4,13]. Patients may experience physical, psychological, and economic burden due to prolonged and recurrent illness and high treatment costs [24,25]. Significant fibrosis is a hallmark of liver disease progression, and early diagnosis guides significance for subsequent clinical treatment decisions. In addition, early and effective intervention for a population that once had chronic schistosomiasis can avoid the occurrence of advanced schistosomiasis, which is not only beneficial to public health service for the residents in endemic areas, but also is of great significance for improving prognosis and survival quality.

A liver biopsy is a special test for diagnosing liver fibrosis, providing auxiliary information for accurate diagnosis and prognosis of liver disease, but its invasiveness makes it limited in clinical applications. In recent years, with the use of non-invasive liver fibrosis markers in the clinical practice, the prediction, and monitoring of chronic liver disease are increasingly independent of liver biopsy. Studies on non-invasive markers of liver fibrosis are seen in other chronic liver diseases. Whereas few studies have been conducted on the diagnosis of hepatic fibrosis in schistosomiasis, it was observed in our follow-up that patients with chronic schistosomiasis could have fibrosis progression after discontinuation of anthelmintic drugs.

In this study, we retrospectively analyzed 36 humoral (serum/plasma) markers and related clinical data of 271 schistosomiasis patients (132 CS cases, 139 AS cases) and found that HGB, MON, GLB, GGT, APTT, VIII, and Fbg were related to fibrosis. The new non-invasive diagnostic function model that we established can predict the possibility of chronic schistosomiasis cases developing into advanced schistosomiasis cases. The model is sufficiently reliable in diagnosing fibrosis and predicting the progression of liver fibrosis.

Previously, it was found that hepatitis B virus infection and schistosomiasis interacted in liver damage [26]. Recent studies have also shown that there was no significant correlation between HBV infection status and the development of CS to AS (ascites type) [27], the correlation analysis of this study further showed that HBV infection in CS cases was not an influencing factor in the development of AS.

There are three markers reflecting coagulation function in the FDA model that was constructed from seven markers in this study. The parasitism and migration of *S. japonicum* in the venous system, and the deposition of eggs in liver tissue causes specific pathological reactions in its definitive hosts [28]. Theoretically, vascular injury first induces a local inflammatory response, followed by an imbalance of the coagulation and fibrinolytic systems, and their interactions ultimately lead to a systemic pathological response in the host. Systemic coagulation, as a follow-up response to inflammation of schistosomes, plays an important role in the compensation of parasitic immunity [29,30,31]. However, to maintain the homeostasis of the blood system, the oversecreted fibrin in coagulation needs to be further degraded by fibrinolytic factors such as plasminogen and plasmin [32]. Previous studies have reported abnormal coagulation function in patients with schistosomiasis, especially in patients with advanced schistosomiasis [29,30,31,32]. If the coagulation and anticoagulation systems, as well as the fibrinolytic and antifibrinolytic systems are balanced, a hypercoagulable state or bleeding tendency may occur, with elevations of VIII mainly seen in hypercoagulable states and thrombotic diseases [33,34,35], this may be due to the deposition of a large number of eggs in the mesenteric blood vessels after infection with schistosomes, and the egg antigen stimulates the blood vessel wall and activates the coagulation system. Currently, there are no clinically useful serum biomarkers or assays to assess fibrosis in patients with advanced schistosomiasis. APTT-activated partial thromboplastin time, Fbg fibrinogen, and factor VIII activity may be good candidates.

In summary, we found that the discriminant function that was established by the use of body fluid (serum/plasma) markers can effectively warn the occurrence of AS. FDA is a reliable prediction model with strong practicability. It plays an important guiding role in effectively controlling the occurrence of AS and lays a solid foundation for achieving the goal of schistosomiasis elimination. We will further carry out early treatment intervention for CS patients who may develop into AS cases, and establish the best treatment strategy for patients with chronic schistosomiasis to effectively prevent the development of AS.

Being an exploratory study with a small sample size, the participants in our study were selected from different schistosomiasis endemic counties (cities, districts) of Jiangxi province which might present a certain representativeness. In addition, a series of independent comparisons were conducted in our study which might increase the statistical error. Thus, further study that is based on an increased sample size or random sampling strategy should be conducted to verify the results that were explored in our study.

## 4. Methods

### 4.1. Patients

The case-control study included two groups of cases from eight counties (Nanchang, Xinjian, Jinxian, Xingzi, Duchang, Yongxiu, Poyang, Yugan) in the schistosomiasis severely endemic area of Poyang Lake District, Jiangxi Province, from February to March 2013. A total of 271 patients were recruited, including 139 AS patients and 132 CS patients who were diagnosed according to the “Diagnostic Criteria for Schistosomiasis” (WS261-2006) that was issued by the Ministry of Health of the People’s Republic of China [36] (Appendix A). These cases exclude diseases such as metabolic hereditary liver disease, other parasitic infections, tumors, cardiovascular system, kidney disease, respiratory system, digestive system, diabetes, infection and tissue necrosis, bacteremia, and systemic lupus erythematosus, while minimizing the confounding effects of other liver diseases (except hepatitis B). In addition, we also retained 109 CS patients to observe whether they had developed into AS by 2020 to evaluate the accuracy of the discriminant function warning.

### 4.2. Blood-Based Biomarkers

For all study subjects, elbow venous blood was collected and sent to the First Affiliated Hospital of Nanchang University within 2 h. Blood routine examinations and biomarkers that were related to liver function, fibrin degradation product D-dimer, coagulation index, HBV, alpha-fetoprotein, and 36 tests of liver fibrosis were conducted (Table 7). Biomarker detection operations were carried out in strict accordance with the instructions.

### 4.3. Data Preprocessing

Among the 36 blood biomarkers, 30 blood biomarkers were continuous variables followed by normality testing. Although the logarithm was taken for correction, there were still some data that were not normally distributed. Therefore, in order to facilitate the unified analysis of the data, we intend to convert these continuous variables into categorical variables. The receiver operating characteristics (ROC) curve was applied to evaluate the various biomarkers in AS and CS patients to find the optimal clinical diagnosis critical value and complete the qualitative classification. The biomarkers were assigned a value as ″0″ when it is less than the critical value, otherwise is ″1″. Biomarkers with area under the ROC curve (AUC) ≤ 0.5, or *p* > 0.05 were excluded.

### 4.4. Statistical Analysis

Statistical analysis was performed using the SPSS Statistics 22.0 software (SPSS Inc., Chicago, IL, USA) with a test level of α = 0.05. Data analysis methods include the following parts:

First, we generally characterized the participants according to gender and age to ensure consistency between the samples. Then, univariate analysis was performed on the differences between the biomarker classification variables of the AS group and the CS group by Chi-square test to select the variables that could be used for analysis in the next step.

We also consider that there may be a certain correlation between variables with no significant difference in univariate analysis and other confounding variables. To avoid the true effect of this variable being masked by the effect of other confounding indicators, all variables with *p* < 0.2 were included in the multivariate analysis [37], and then variables with *p* < 0.10 remained in the multivariate model after the stepwise backward selection process. The final report is presented as odds ratios (ORs) with 95% confidence intervals (95% CI) and significance levels (*p*-values).

Fisher discriminant analysis (FDA) is a commonly used multivariate statistical method [38], which uses projection techniques for dimensionality reduction to determine a linear function of variables to maximize differences between the samples of multiple classes and minimize the differences between the samples of the same class [39]. Therefore, we use the variables that were screened out by multivariate analysis to establish the discriminant function by using the FDA model selection step method and conduct a self-test on the established discriminant function.

## Figures and Tables

**Table 1 pathogens-11-01004-t001:** Basic information of the patients.

Variable	AS			CS		
	Male	Female	Total	Male	Female	Total
Number	79	60	139	65	67	132
Age, years						
Mean ± SD	64.1 ± 9.7	57.8 ± 10.1	61.4 ± 10.3	61.8 ± 8.5	59.9 ± 9.2	60.8 ± 8.9
Range	28–82	33–76	28–82	26–81	32–78	26–81

AS, advanced schistosomiasis; CS, chronic schistosomiasis.

**Table 2 pathogens-11-01004-t002:** Optimal diagnostic cut-off of blood biomarkers for screening AS and CS patients by ROC.

Biomarkers (Unit)	Critical Value	ROC			
		AUC	Std. Error	95% CI	*p*-Value
WBC (10^9^/L) *	8.82	0.450	0.035	0.381–0.518	0.154
RBC (10^12^/L)	4.985	0.654	0.035	0.586–0.722	<0.001
HGB (g/L)	128.5	0.610	0.036	0.541–0.681	0.003
PLT (10^9^/L) *	205.0	0.547	0.035	0.478–0.616	0.184
LYM (%)	34.85	0.612	0.034	0.545–0.679	0.001
MON (%)	7.95	0.673	0.033	0.608–0.738	<0.001
NEU (%)	54.75	0.768	0.032	0.705–0.832	<0.001
EOS (%) *	1.35	0.512	0.035	0.443–0.581	0.733
BAS (%)	0.55	0.636	0.034	0.570–0.702	<0.001
ALT (U/L) *	15.5	0.554	0.035	0.485–0.623	0.126
AST (μ/L)	38.5	0.649	0.033	0.584–0.714	<0.001
TBiL (μmol/L)	12.5	0.619	0.034	0.553–0.686	0.001
DbiL (μmol/L)	5.25	0.634	0.034	0.568–0.700	<0.001
TP (g/L) *	76.3	0.523	0.035	0.455–0.592	0.504
ALB (g/L) *	53.9	0.269	0.031	0.209–0.330	<0.001
GLB (g/L)	30.5	0.732	0.030	0.672–0.792	<0.001
ALB/GLB	1.545	0.840	0.026	0.790–0.890	<0.001
GGT (U/L)	28.5	0.653	0.033	0.588–0.718	<0.001
ALP (U/L)	94.5	0.655	0.033	0.591–0.720	<0.001
D-dimer (mg/L)	0.51	0.703	0.031	0.642–0.765	<0.001
PT (s)	11.15	0.701	0.031	0.639–0.763	<0.001
APTT (s)	26.35	0.695	0.032	0.633–0.757	<0.001
Fbg (g/L)	2.03	0.614	0.038	0.540–0.688	0.002
TT (s)	18.05	0.676	0.032	0.613–0.740	<0.001
VⅢ (%)	134.25	0.572	0.038	0.498–0.646	0.049
AFP (ng/mL) *	1.585	0.531	0.035	0.462–0.601	0.371
HA (ng/mL)	131.92	0.601	0.034	0.534–0.669	0.004
PCⅢ (ng/mL)	109.7	0.597	0.034	0.530–0.664	0.006
IV-C (ng/mL) *	124.23	0.512	0.035	0.442–0.581	0.738
LN (ng/mL)	151.555	0.609	0.034	0.542–0.676	0.002

ROC, receiver operating characteristics; AUC, area under curve; AS, advanced schistosomiasis; CS, chronic schistosomiasis * Abandoned.

**Table 3 pathogens-11-01004-t003:** Univariate analysis of qualitative blood biomarkers in the AS and CS groups.

Biomarkers	χ^2^	OR	95% CI	*p*-Value
SJ *	0.026	0.956	0.556–1.644	0.872
RBC	2.489	0.451	0.164–1.239	0.115
HGB	12.677	0.413	0.253–0.675	<0.001
LYM	21.007	3.903	2.134–7.137	<0.001
MON	38.155	6.116	3.328–11.239	<0.001
NEU	29.009	0.215	0.120–0.385	<0.001
BAS	17.648	3.079	1.804–5.254	<0.001
AST	16.037	2.862	1.697–4.825	<0.001
TBiL	11.532	2.322	1.423–3.790	0.001
DBiL	20.160	3.380	1.962–5.820	<0.001
GLB	50.025	8.551	4.481–16.316	<0.001
ALB/GLB	48.222	0.148	0.084–0.260	<0.001
GGT	18.757	2.943	1.795–4.825	<0.001
ALP	16.692	3.013	1.758–5.164	<0.001
D-dimer	31.279	4.166	2.499–6.947	<0.001
PT	37.575	5.894	3.240–10.719	<0.001
APTT	29.340	4.276	2.488–7.347	<0.001
Fbg	26.947	0.109	0.041–0.286	<0.001
TT	27.726	5.618	2.826–11.168	<0.001
VⅢ	11.091	0.434	0.265–0.712	0.001
HA	9.547	2.192	1.327–3.620	0.002
PCⅢ	18.835	4.958	2.286–10.753	<0.001
LN	15.671	3.238	1.781–5.887	<0.001
HBsAg *	0.045	0.937	0.516–1.702	0.832
HBsAb *	0.269	1.137	0.699–1.849	0.604
HBeAg *	0.953	0.000	0.000–0.000	0.329
HBeAb *	0.866	1.284	0.758–2.173	0.352
HBcAb	3.395	1.576	0.970–2.560	0.065

AS, advanced schistosomiasis; CS, chronic schistosomiasis * Abandoned.

**Table 4 pathogens-11-01004-t004:** Multivariate analysis of qualitative blood biomarkers in the AS and CS groups.

Biomarkers	OR	95% CI	*p*-Value
HGB	0.328	0.170–0.631	0.001
LYM	2.665	1.200–5.916	0.016
MON	3.424	1.541–7.606	0.003
DBiL	2.497	1.179–5.286	0.017
GLB	5.262	2.373–11.671	<0.001
GGT	2.723	1.408–5.266	0.003
APTT	3.641	1.837–7.216	<0.001
Fbg	0.230	0.073–0.728	0.012
VIII	0.498	0.258–0.961	0.038

AS, advanced schistosomiasis; CS, chronic schistosomiasis.

**Table 5 pathogens-11-01004-t005:** Classification results of the original and cross-validated methods.

Method	Actual	Predicted (*n*, %)		Consistency Rates (%)
		CS	AS	
Cross-validated	CS	117 (88.6)	15 (11.4)	86.7
	AS	21 (15.1)	118 (84.9)	

AS, advanced schistosomiasis; CS, chronic schistosomiasis.

**Table 6 pathogens-11-01004-t006:** Results of the 109 CS patients at the return workup.

Predicted	Number	The Lost Number	Pay a Return Visit (*n*, %)		Consistency Rates (%)
			CS	AS	
CS	75	2	65 (89.0)	8 (11.0)	81.4
AS	34	5	11 (37.9)	18 (62.1)	

AS, advanced schistosomiasis; CS, chronic schistosomiasis.

**Table 7 pathogens-11-01004-t007:** Blood biomarker test methods and product providers.

Test Items	Test Method	Product Provider
Schistosomiasis japonica, SJ	ELISA	Shenzhen HuaKang Co., Ltd., China
White blood cells, WBC	Flow cytometer	sysmex XE-2100, Japan
Red blood cells, RBC	Flow cytometer	sysmex XE-2100, Japan
Hemoglobin, HGB	Flow cytometer	sysmex XE-2100, Japan
Platelets, PLT	Flow cytometer	sysmex XE-2100, Japan
Lymphocyte, LYM	Flow cytometer	sysmex XE-2100, Japan
Monocyte, MON	Flow cytometer	sysmex XE-2100, Japan
Neutrophil, NEU	Flow cytometer	sysmex XE-2100, Japan
Eosinophil, EOS	Flow cytometer	sysmex XE-2100, Japan
Basophil, BAS	Flow cytometer	sysmex XE-2100, Japan
Alanine aminotransferase, ALT	Rate method	Roche, Basel, Swiss
Aspartate aminotransferase, AST	Rate method	Roche, Basel, Swiss
Total bilirubin, TBiL	Endpoint method	Roche, Basel, Swiss
Direct bilirubin, DBiL	Endpoint method	Roche, Basel, Swiss
Total protein, TP	Endpoint method	Roche, Basel, Swiss
Albumin, ALB	Endpoint method	Roche, Basel, Swiss
Globulin, GLB	Endpoint method	Roche, Basel, Swiss
Albumin/globulin ratio, ALB/GLB		
γ-Glutamyltransferase, GGT	Rate method	Roche, Basel, Swiss
Alkaline phosphatase, ALP	Rate method	Roche, Basel, Swiss
D-dimer, D-D	Immunoturbidimetric method	Siemens Healthcare GmbH, Munich, Germany
Prothrombin time, PT	Colorimetry	Roche, Basel, Swiss
Activated partial thromboplastin time, APTT	Colorimetry	Roche, Basel, Swiss
Fibrinogen, Fbg	Class	Roche, Basel, Swiss
Thrombin time, TT	Colorimetry	Roche, Basel, Swiss
Coagulation factor VⅢ, VⅢ	Colorimetry	Roche, Basel, Swiss
HBsAg	ELISA	Abbott, North Chicago, Illinois, USA
HBsAb	ELISA	Abbott, North Chicago, Illinois, USA
HBeAg	ELISA	Abbott, North Chicago, Illinois, USA
HBeAb	ELISA	Abbott, North Chicago, Illinois, USA
HBcAb	ELISA	Abbott, North Chicago, Illinois, USA
α-fetoprotein, AFP	RIA	Roche, Basel, Swiss
Hyaluronic acid, HA	RIA	Shanghai Navy Medical Institution, Shanghai, China
Procollagen type Ⅲ, PCⅢ	RIA	Shanghai Navy Medical Institution, Shanghai, China
Type IV collagen, IV-C	RIA	Shanghai Navy Medical Institution, Shanghai, China
Laminin, LN	RIA	Shanghai Navy Medical Institution, Shanghai, China

## Data Availability

The data that support the figures within this paper and other findings of this study are available from the corresponding authors upon reasonable request.

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
