# Peer review of "Predicting the Occurrence of Advanced Schistosomiasis Based on FISHER Discriminant Analysis of Hematological Biomarkers"

_pathogens, 2022, doi:10.3390/pathogens11091004_

Round 1

Reviewer 1 Report

As mentioned by the authors “Liver biopsy is a special test for diagnosing liver fibrosis, providing auxiliary information for accurate diagnosis and prognosis of liver disease, but its invasiveness makes it limited in clinical application”

Research that reduces invasive diagnostic methods is important

I suggest making it clearer in the introduction the reasons for choosing biomarkers

Author Response

Dear Reviewers:

Thank you for your interest in our manuscript entitled “Predicting the occurrence of advanced schistosomiasis based on Fisher discriminant analysis of hematological biomarkers”(pathogens-1813889). We have studies comments carefully and have made correction which we hope meet the final approval. The alterations in the text are highlighted in the paper as requested. The main corrections in the paper and the responses to the reviewer’s comments are as following:

  1. As mentioned by the authors “Liver biopsy is a special test for diagnosing liver fibrosis, providing auxiliary information for accurate diagnosis and prognosis of liver disease, but its invasiveness makes it limited in clinical application”. Research that reduces invasive diagnostic methods is important. I suggest making it clearer in the introduction the reasons for choosing biomarkers.

Response: Thank you very much for your comments. The reason for choosing biomarkers had been added in the third paragraph of introduction.

In addition,We have invited experts in this field helping us improved the language and modified the inappropriate expression in our resubmitted manuscript.

We tried our best to improve the manuscript and made some changes in the manuscript. These changes will not influence the content and framework of the paper. And here we did not list the changes but marked up using the“track changes”in revised paper.

We appreciate for Editors/Reviewers’ warm work earnestly, and hope that the correction will meet with approval.

Once again, thank you very much for your comments and suggestions.

Reviewer 2 Report

Thank you for sharing the article on the prediction of an advanced status of schistosomiasis based on blood biomarkers. Here some minor issues that could help to improve the article. 

L: Here you mention chronic schistosomiasis and advanced schistosomiasis. Please clarify if the terms differ and if so how or whether they mean the same. 

L84: Do AS and CS mean advanced and chronic schistosomiasis? Please explain those abbreviations. As you stratify your participants by AS and CS, please come up with a clear definition for both terms. In L209 ff you list general exclusion criteria, but you don't define inclusion criteria differentiating AS from CS. L195-196 despite chronic schistosomiasis seems to develop to advanced schistosomiasis, a clear definition and distinction of both terms are needed. 

Author Response

Dear Reviewers:

Thank you for your interest in our manuscript entitled “Predicting the occurrence of advanced schistosomiasis based on Fisher discriminant analysis of hematological biomarkers”(pathogens-1813889). We have studies comments carefully and have made correction which we hope meet the final approval. The alterations in the text are highlighted in the paper as requested. The main corrections in the paper and the responses to the reviewer’s comments are as following:

  1. Here you mention chronic schistosomiasis and advanced schistosomiasis. Please clarify if the terms differ and if so how or whether they mean the same.

Response: Thanks a lot for your comments. Chronic schistosomiasis and advanced schistosomiasis are different clinic stages of schistosomiasis japonica with different symptoms and signs according to the Diagnostic Criteria for Schistosomiasis(WS 261-2006) issued by the Ministry of Health of the People's Republic of China. Chronic schistosomiasis will occur when people are exposed to infested water with cercaria of schistosome repeatedly or are infected by small dose of cercaria several times or frequently in daily life. There are slight or no obvious symptoms or signs in most of patients with chronic schistosomiasis. A few individuals with chronic schistosomiasis develop periportal fibrosis of the liver, which may result in the hepatosplenic form of the disease.

Advanced schistosomiasis is one form of chronic schistosomiasis, resulting from repeated exposure to schistosome cercariae or one infection with a large quantity of cercariae but lack of timely and effectively schistosommicidal treatment. Advanced schistosomiasis mainly manifests as portal hypertension, gastrointestinal bleeding and hepatosplenmegaly. Owing to the complexity of symptoms, advanced schistosomiasis japonica was classified into four clinical subtypes in China: ascites, megalosplenia, colonic tumoroid proliferation, and dwarfism.

We had added a description of the clinical manifestations of chronic schistosomiasis in the second paragraph of introduction, as follows: Clinical presentation varies in chronic schistosomiasis cases, but many cases present mild symptoms or absence of symptoms thus leading to misdiagnosis or lack of treatment.

  1. L84: Do AS and CS mean advanced and chronic schistosomiasis? Please explain those abbreviations. As you stratify your participants by AS and CS, please come up with a clear definition for both terms. In L209 ff you list general exclusion criteria, but you don't define inclusion criteria differentiating AS from CS. L195-196 despite chronic schistosomiasis seems to develop to advanced schistosomiasis, a clear definition and distinction of both terms are needed.

Response: Thank you very much for your comments. Yes, AS is the acronym of advanced schistosomiasis and CS is the acronym of chronic schistosomiasis. The definition of chronic schistosomiasis and advanced schistosomiasis were represented in response to Comment 1. We stratified the participants according to the Diagnostic Criteria for Schistosomiasis(WS 261-2006), the detailed terms of criteria will upload as supplementary material. The aim of the exclusion of other diseases in L209 ff was to exclude the interference factors from other liver disorders diseases instead of exclusion criteria of AS or CS. 

In addition,We have invited experts in this field helping us improved the language and modified the inappropriate expression in our resubmitted manuscript.

We tried our best to improve the manuscript and made some changes in the manuscript. These changes will not influence the content and framework of the paper. And here we did not list the changes but marked up using the“track changes”in revised paper.

We appreciate for Editors/Reviewers’ warm work earnestly, and hope that the correction will meet with approval.

Once again, thank you very much for your comments and suggestions.

Reviewer 3 Report

Estimated Authors,

thank you for the opportunity to review this very interesting paper. In this retrospective study, Hu et al retrospectively analyzed 36 humoral (serum/plasma) markers and related clinical data of 271 schistosomiasis patients (132 CS cases, 139 AS cases) and found that HGB, MON, GLB, GGT, APTT, VIII, and Fbg were related to fibrosis.

From the point of view of the present reviewer, study design and data reporting are accurate and well written: I've no specific requests about it. However, I've noticed a series of potential shortcomings that should be addressed before the eventual acceptance of this paper, and more precisely:

1) Authors have performed a series of independent comparisons; as a consequence, the risk for increased alpha error increases accordingly and should be addressed at least in the discussion;

2) a lot of items are assessed and included in the eventual analyses, but the meaning of the acronyms is reported in a later section of the paper; please include their meanings in the caption of the tables. Similarly, the meaning of AS and CS should be provided across the text and tables, and not only in the abstract.

3) please include as a supplementary materials the individuals ROC curves;

4) please note that a p value equals to 0.000 does not exist. It is a common mistake from SPSS, and therefore please fix it accordingly, as p < 0.001.

5) eventually, please explain the rationale of the equations in section 2.5 by taking in account that, being the paper reporting methods section  in later pages, readers could understand it only afterwards (in other words, I would suggest to share some explanation about these equations even in the results section).

6) eventually, in the limitations section authors should discuss whether the sample size may be acknowledged as representative or not. In later case, the study will retain its significant but as an exploratory research.

Author Response

Dear Reviewers:

Thank you for your interest in our manuscript entitled “Predicting the occurrence of advanced schistosomiasis based on Fisher discriminant analysis of hematological biomarkers”(pathogens-1813889). We have studies comments carefully and have made correction which we hope meet the final approval. The alterations in the text are highlighted in the paper as requested. The main corrections in the paper and the responses to the reviewer’s comments are as following:

  1. Authors have performed a series of independent comparisons; as a consequence, the risk for increased alpha error increases accordingly and should be addressed at least in the discussion.

Response: Thanks a lot for reminding us of this important point. Yes, the risk of alpha error increased for performed a series of independent comparisons. The effective way to decrease the risk is to enlarge the size of sample. But, the size of sample in this study could not be enlarged due to the difficulties in recruiting people. In order to improve representativeness of the sample, the participants were selected from different schistosomiasis endemic counties (cities, districts) of Jiangxi province. In addition, all variables with P value <0.2 in the univariate analysis were all included in multivariate analysis to avoid losing potential options and increase the credibility of the results. We also discussed the solutions in the discussion section as followings:

“Being an exploratory study with a small sample size, the participants in our study were selected from different schistosomiasis endemic counties (cities, districts) of Jiangxi province which might present a certain representativeness. In addition, a series of independent comparisons were conducted in our study which might increase statistical error. Thus further study based on increased sample size or random sampling strategy should be conducted to verify the results explored in our study.”

  1. a lot of items are assessed and included in the eventual analyses, but the meaning of the acronyms is reported in a later section of the paper; please include their meanings in the caption of the tables. Similarly, the meaning of AS and CS should be provided across the text and tables, and not only in the abstract.

Response: Thanks a lot for reminding us of this important point. We have list the meanings of all abbreviations and acronyms at the end of the summary, as follows:

Abbreviations and Acronyms: CS, chronic schistosomiasis; AS, advanced schistosomiasis; FDA, fisher discriminant analysis; ROC, receiver operating characteristics; AUC, area under curve; SJ, schistosomiasis japonica; WBC, white blood cells; RBC, red blood cells; HGB, hemoglobin; PLT, platelets; LYM, lymphocyte; MON, monocyte; NEU, neutrophil; EOS, eosinophil; BAS, basophil; ALT, alanine aminotransferase; AST, aspartate aminotransferase; TBiL, total bilirubin; DBiL, direct bilirubin; TP, total protein; ALB, Albumin; GLB, Globulin; ALB/GLB, albumin/globulin ratio; GGT, γ-Glutamyltransferase; ALP, alkaline phosphatase; D-D, D-dimer; PT, prothrombin time; APTT, activated partial thromboplastin time; Fbg, fibrinogen; TT, thrombin time; VⅢ, coagulation factor VⅢ; AFP, α-fetoprotein; HA, Hyaluronic acid; PCⅢ, procollagen type Ⅲ; IV-C, type IV collagen; LN, laminin

  1. Please include as a supplementary materials the individuals ROC curves.

Response: Thanks a lot for reminding us of this important point. We had made the individuals ROC curves and upload as supplementary material. It’s shown as follows:

  1. Please note that a p value equals to 0.000 does not exist. It is a common mistake from SPSS, and therefore please fix it accordingly, as p < 0.001.

Response: Thanks a lot for reminding us of this important point. We have revised these errors throughout the text.

  1. Eventually, please explain the rationale of the equations in section 2.5 by taking in account that, being the paper reporting methods section in later pages, readers could understand it only afterwards (in other words, I would suggest to share some explanation about these equations even in the results section).

Response: Your comment is very constructive and we have added explanations about these equations in the results section.

  1. Eventually, in the limitations section authors should discuss whether the sample size may be acknowledged as representative or not. In later case, the study will retain its significant but as an exploratory research.

Response: Thank you very much for your suggestion. We've cleared this issue in the limitations section.

In addition,We have invited experts in this field helping us improved the language and modified the inappropriate expression in our resubmitted manuscript.

We tried our best to improve the manuscript and made some changes in the manuscript. These changes will not influence the content and framework of the paper. And here we did not list the changes but marked up using the“track changes”in revised paper.

We appreciate for Editors/Reviewers’ warm work earnestly, and hope that the correction will meet with approval.

Once again, thank you very much for your comments and suggestions.
